# Sustainable Economic Growth and FDI Inflow: A Comparative Panel Econometric Analysis of Low-Income and Middle-Income Nations

**Mohammad Anamul Haque** [1,*] ⓘ**, Syed Mehmood Raza Shah** [2] **and Muhammad Usman Arshad** [3] ⓘ

1   School of Finance, Central University of Finance and Economics, Beijing 100081, China
2   School of Finance, Dongbei University of Finance and Economics, Dalian 116025, China
3   Department of Commerce, University of Gujrat, Gujrat 50700, Pakistan
*   Correspondence: anamulhaque81@yahoo.com

**Abstract:** The study examines the effect of sustainable economic growth on "FDI inflow" using comparative panel econometrics on two panels: "low-income" and "middle-income" economies between 1970 and 2021. For this, 18 "low-income" and 53 "middle-income" economies constitute the sample. The data were retrieved from the "world development indicator" website. Pre-diagnostic and post-diagnostic estimations were performed using static panel and dynamic panel approaches. Sustainable growth increases "FDI inflow" in "low-income" and "middle-income" economies during the study period, according to the findings. In addition, trade openness and the exchange rate have the potential to boost "FDI inflow" in "low-income" economies. Similarly, in "middle-income" economies, the real growth rate and exchange rate are significant boosts, however inflation significantly reduces the "FDI inflow". The findings show that policymakers in "low-income" and "middle-income" economies should maintain long-term, sustainable economic growth in order to attract more "FDI inflow" in their respective economies. Compared to the current state of knowledge in the subject, the study's findings provide evidence for "low-income" and "middle-income" nations that have been mainly overlooked in terms of sustainable growth for attracting FDI inflow. The study's outcomes are applicable and generalizable only for "middle-income" and "low-income" economies. Future researchers may include additional control factors and expand the scope of the study to include "high-income" groups.

**Keywords:** sustainable economic growth; FDI inflow; trade openness; real growth; inflation; exchange rate



## 1. Introduction

FDI net inflows refer to the volume of inward capital inflows created by foreign investors in the host country, such as capital invested plus interpersonal and inter loans [1]. FDI has evolved into a significant source of independent financing for developing regions [2]. It differs from other types of external private investment because financiers expect long-term profits from manufacturing activities they control. FDI increases employment as companies establish new businesses abroad [3]. Residents may acquire greater income and purchasing power, hence supporting broader economic expansion in the designated nations [4]. Figure 1 indicates the trend line of average FDI as a percentage of GDP in "low-income" and "middle-income" economies from the period 1970–2021. The lowest point of "FDI inflow" in "low-income" economies was in 1984–85, after which it rose until 2010 and then dropped until 2020. Similarly, the "FDI inflow" in "middle-income" economies was lowest in 1988–1989, then increased until 2008–2009, when it started to drop again until 2020.

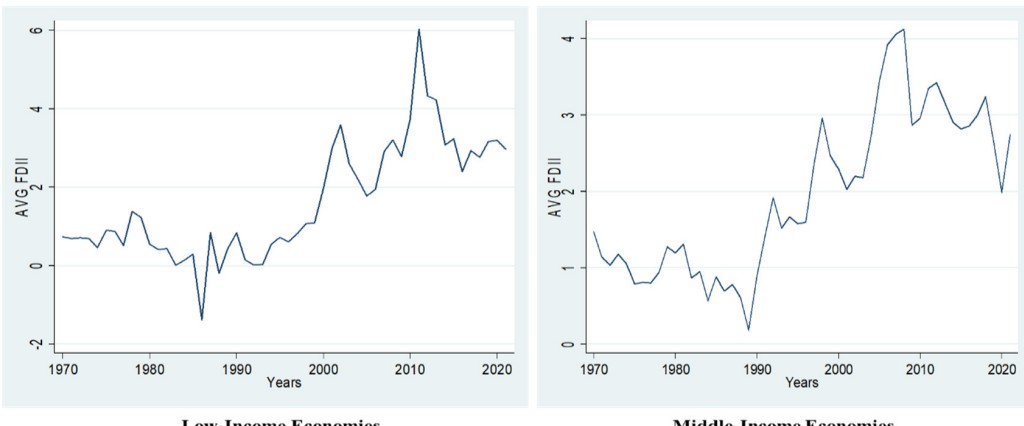

**Figure 1.** Average FDI net inflow (% of GDP).

People living in poverty benefit from long-term sustainable economic development. Energy conservation and public transit expansion reduce air pollution, which helps asthma and heart disease. Homes and businesses that are efficient would be safer and more inviting. Large markets, political and macroeconomic stability, GDP, the regulatory environment, and the repatriation of earnings influence FDI. Sustainable economic growth ensures the long-term expansion of the economy as a whole. The trend line of sustainable economic growth as assessed by per capita income from 1970 to 2021 is depicted in Figure 2. Figure 2 depicts a decline in sustainable economic growth in "low-income" economies from 1970 to 1994, followed by an increase to the present day. However, the "middle-income" economies have seen an upward tendency since 1970. It deduced that "middle-income" economies have long-term sustainable economic growth, while "low-income" economies have short-term sustainable economic growth, as shown in Figure 2.

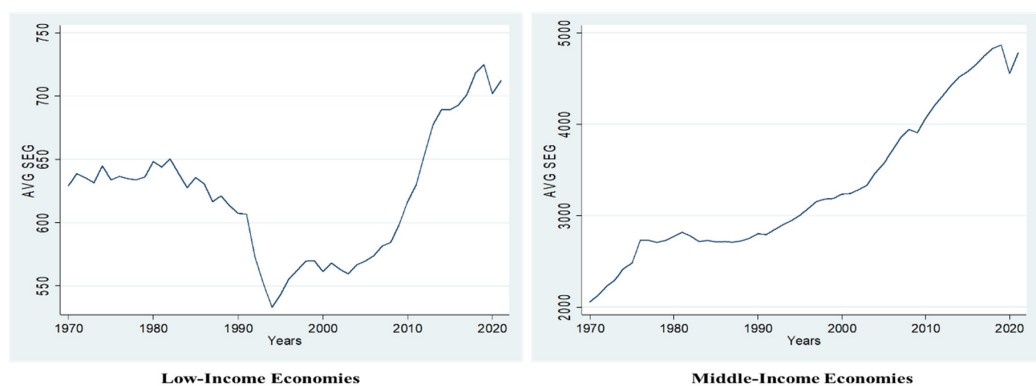

**Figure 2.** Average Sustainable Economic Growth.

This study primarily focuses on the effect of sustainable economic growth on "FDI inflow" for "low-income" and "middle-income" economies from 1970 to 2021. In addition, the study evaluates the effects of trade openness, real growth, exchange rate, and inflation on the target population within the same time period. The following are the precise research aims of this study:

1. To explore the impact of sustainable economic growth on "FDI inflow" for "low-income" and "middle-income" economies.
2. To examine the effect of trade openness, real growth, exchange rate, and inflation on "FDI inflow" for the target population.

"FDI inflow" refers to the money provided by foreign investors in any country. "FDI inflow" helps to boost an economy's foreign reserves. Stable, steady, and long-term economic growth can play a crucial role in increasing the level of "FDI inflow" in a host

country. The government and other important stakeholders must maintain sustainable economic growth in order to continue to attract "FDI inflow". The findings help "low-income" and "middle-income" economies attain their required "FDI inflow" utilizing sustainable economic growth.

Additionally, the study includes the literature and hypothesis in Section 2, methodology in Section 3, analysis, interpretation, and discussion in Section 4, and conclusion with recommendations in Section 5.

## 2. Literature Review

Sustainable economic growth is defined as a pace of growth that can be sustained over the long term, especially for the next generations [5]. There is unquestionably a relationship between current economic growth and anticipated economic growth [6]. Trade openness is calculated by summing a country's imports and imports as a proportion of its gross domestic product [7]. The real rate of economic growth measures the annual change or expansion of a country's gross domestic product [8]. GDP measures the total market value of all goods and services produced by a country during a certain time period [9]. "Official exchange rate" refers to the exchange rate set by the country's authorities or the market rate recognized by law [4]. Using monthly values, the annual average is calculated [10]. Inflation refers to the rate of increase in prices over a specified time period [11]. Inflation is frequently measured in broad terms, such as the general increase in prices or the increase in a country's cost of living [12]. The theory that discusses the relationship between "FDI inflow" and sustainable economic growth is named as the growth theory of FDI that was developed by [13]. The theory indicates that a long-run economic growth rate enhances the confidence of foreign investors in investing in a host nation. The growth theory of "FDI inflow" forms a positive relationship between sustainable economic growth and "FDI inflow" [14].

Ref [15] found a positive and significant impact on long-run economic growth and "FDI inflow" in the case of Saudi Arabia. Additionally, [16] found a bidirectional causal relationship between long-run economic growth and "FDI inflow" in the case of the MENA region. Furthermore, [17] also found a positive and potential impact of long-run economic growth on "FDI inflow" in the case of Sri Lanka. However, [18] confirmed bidirectional causality between long-run economic growth and "FDI inflow" in the case of the Caribbean region. Similarly, [7] found a long-run positive impact of sustainable economic growth on "FDI inflow" for Pakistan. Furthermore, [19] found a positive yet significant association between sustainable economic growth and "FDI inflow" in the case of 57 developing nations. The positive role of sustainable economic growth strongly enhances the "FDI inflow" in different regions of the world as seen in the past literature such as [2] in eastern and central European countries, [3] in China, [14] in the case of Nigeria, [20] in the case of west Africa, [21] in the case of China again, [22] using a panel of 10 selected "low-income" countries, [6] in the case of Sub-Saharan Africa, [23] in the case of Rwanda, [24] in the case of African countries, [25] in the case of Oman, [26] in the case of 21 selected Asian countries, [27] in the case of Estonia, and, finally [28] for a panel of ECO country members. As none of the study considered a comparative panel of "low-income" and "middle-income" countries to examine the impact of sustainable economic growth for attracting FDI inflow, therefore, the present study needs to evaluate the research gap in this domain.

The positive link between sustainable economic growth and "FDI inflow" requires the formation of a first hypothesis that follows as the primary objective of the study.

**H₁.** *Sustainable economic growth significantly enhances the "FDI inflow".*

The past literature provides a large number of evidence in support of a positive and highly significant relationship between trade openness and "FDI inflow" in different regions of the world using time series, as well using aggregate panels of different economies such as in [11], which uses a comparative panel of developed and developing countries; [29] in the case of post-communist economies; [7] in the case of Pakistan; [10] for a panel of selected

ASEAN countries; [18] in the case of a selected panel from Caribbean countries; [30] in the case of India; [31] in the case of India again; and, finally [32] in the case of some countries from the MENA region. However, some important studies still provided the insignificant yet positive impact of trade openness on "FDI inflow" for individual countries as well as on some panel studies such as [12] in the case of Africa; [33] for a panel of emerging markets; [34] in the case of developing and emerging countries; [35] in the case of Saudi Arabia; and [36] in the case of some of the MENA countries. The majority of research studies as per the literature analysis revealed a lack of comparative analysis between "low-income" and "middle-income" countries, therefore, the study requires an analysis of this research gap. The second hypothesis was formed to test the positive impact of "FDI inflow" based on the majority of findings from the past literature as follows as the secondary objective.

**H₂.** *Trade openness significantly enhances the "FDI inflow".*

Many evidence in the past provides a positive impact of real growth rate on "FDI inflow" for different regions, countries, and panels such as [11] in the case of developing and developed nations; [29] in the case of Vietnam; [37] in the case of the Balkans; [9] in the case of a panel of 37 agriculture based countries; and, finally, [38] in the case of the Sub-Saharan Africa region. However, some important evidence in the literature could not provide the significant impact of the real growth rate for "FDI inflow" in the case of different panels, regions, or countries such as [12] in the case of Africa; [39] in the case of a panel of developing countries; [4] in the case of a panel of seven selection countries from the ASEAN region; [40] in the case of a panel of "middle-income" economies; and, finally, [41] in the case of a panel of developing countries. The critical analysis of the literature evidence revealed that a comparative analysis of "middle-income" and "low-income" nations is lacking. Therefore, the present research entails this gap as a secondary aim of the study. The third hypothesis was formed to test a positive yet significant impact of the real growth rate on "FDI inflow" based on a majority of evidence as the secondary objective.

**H₃.** *Real growth rate significantly increases "FDI inflow".*

The past literature in this domain provides a large number of evidence in support of the positive and significant relationship between the exchange rate and "FDI inflow" for different panels such as [11] for developing countries' panels; [29] in the case of post-communist regions; [10] in the case of a panel of the ASEAN region; and, finally, [42] in the case of India. However, some of the research studies in the past could not provide a significant impact of the exchange rate on "FDI inflow" for different panels, countries, or regions such as [12] in the case of Africa; [39] in the case of developing economies; [8] in the case of developing nations again; and, finally, [42] in the case of India. The analysis of the historical literature evidence revealed that a comparative analysis between "low-income" and "middle-income" countries in the present domain is missing. Therefore, the study must evaluate the impact of real growth rate as the control variable for explaining the FDI inflow. The study forms the fourth hypothesis based on a majority of evidence in support of the positive and significant impact of the exchange rate on "FDI inflow" as the secondary objective.

**H₄.** *Real official exchange rate strongly increases the "FDI inflow".*

Finally, a large number of studies reported a negative and statistically strong impact of inflation on "FDI inflow" in different time series and panel data analyses such as [11] for developing countries' panels; [29] in the case of post-communist regions; [7] in the case of Pakistan; [4] in the case of a panel of seven selected countries from the ASEAN region; and, finally, [42] in the case of India. However, an important study could not provide the significant impact of inflation on "FDI inflow" such as [10] in the case of a panel of some of the ASEAN region countries. The critical analysis of the literature evidence revealed that a comparative analysis of "middle-income" and "low-income" nations is lacking. Therefore, the present research entails this gap as a secondary aim of the study.

The study forms the fifth hypothesis based on the majority of evidence in support of the negative and statistically significant impact of the inflation rate on "FDI inflow" as the secondary objective.

**H₅.** *Inflation rate significantly decreases the "FDI inflow".*

### 3. Data and Methodology

The present research aimed to investigate the impact of sustainable economic growth on "FDI inflow" for "low-income" and "middle-income" economies. To achieve this research objective, the study requires the collection of two separate panels: one for "low-income" economies and the other for "middle-income" economies. The data for this purpose were obtained from the World Bank's data sources, with world development indicators (WDI) using the annual frequency of 1970–2021. The complete specification of the database is www.databank.worldbank.org/source/world-developement-indicators (accessed on 1 March 2022). The study requires a maximum number of countries for data collection purposes and the sample approach is based on convenience sampling that requires the availability and easy access of required data. However, the world development indicators website has available data for 18 "low-income" and 53 "middle-income" countries. Therefore, the final sample includes the data as per available number of countries under "low-income" and "middle income" from WDI for achieving the research objective. The complete description of countries included in the sample is available in Appendix A. The dependent variable of the study is "FDI inflow", while the main independent variable of the study is sustainable economic growth, which is measured as per capita GDP by considering 2015 as a constant. The main difference between economic growth and sustainable economic growth is that the former indicates at what level an economy is able to enhance the production of goods and services for the satisfaction of human needs. However, the latter requires a nation to sustain its natural resources for its future generations along with economic growth. The per capita GDP is related with an economy's positive outlook in terms of better life satisfaction, better health condition, more safety, more education, and better sustainability of natural resources for future generations, therefore, it is a good measure of sustainable economic growth [6,25,43].

Additionally, the study also considered trade openness, exchange rate, inflation rate, and real growth rate as the control variables. As the real growth adjusts the inflation for the calculation of an economy's growth in real terms, therefore, it is a good measure of the growth rate. The detailed measurements of all the variables, including their data sources, and the literature references are summarized in Table 1 below.

**Table 1.** Variable Measurement and Sources.

| Variables | Measurements | Data Sources | References |
|---|---|---|---|
| "FDI inflow" (FDII) | Net Inflow of FDI (% of GDP) | WDI | [6,27,28,31] |
| Sustainable Economic Growth (SEG) | GDP per capita (Constant 2015 USD) | WDI | [6,31,43] |
| Trade Openness (TROP) | Trade as Percentage of GDP | WDI | [34,43] |
| Real Growth (RG) | GDP growth rate (annual Percentage) | WDI | [22,34,41,43] |
| Exchange Rate (EXC) | Official Real Exchange Rate (LCU Per US%, Period Average) | WDI | [6,23] |
| Inflation rate (INFR) | GDP deflator | WDI | [23] |

The study must compare two panels econometrically: "low-income" economies and "middle-income" economies. The data ranges from 1970–2021 on annual frequency. [44] suggested using panel data as a more advantageous method of estimation as compared to other forms of data: time series or cross-sectional. The cross-sectional and time series data have some limitations and issues: normality, multicollinearity, and serial correlation. The panel data, however, are more reliable due to the specific nature of a larger number of observations that enhance normality and control the multicollinearity and serial correlation issues. The panel data estimations require several data analysis procedures: comparative descriptive statistics, comparative unit root testing, and panel regression estimations such

as fixed, random, and pooled OLS estimations. The panel regression requires deciding between fixed effect or random effect using the Hausman specification test. The significance of which confirms the estimations using the fixed-effect model. The confirmation of the fixed-effect model requires testing some assumptions, autocorrelation, heteroscedasticity, and cross-sectional dependency. The violation of serial correlation assumptions requires the use of feasible generalized least square (FGLS) if T > N. However, the panel-corrected standard error (PCSE) may also be used if T < N. Similarly, the violation of the heteroscedastic assumption requires the use of robust estimates for a fixed effect. Furthermore, the violation of cross-sectional dependence requires the use of Driscoll and Kraay standard error (DKSE). Finally, the panel data are also required to estimate the model using the dynamic panel data technique of GMM on the assumption of the endogeneity in the model.

The study aims to explore the Impact of sustainable economic growth on "FDI inflow" for "low-income" and "middle-income" economies during the period of study. The basic economic model of study is as follows.

$$\text{"FDI inflow"} = f \text{ (Sustainable Economic Growth)} \tag{1}$$

The basic econometric model of the study is as follows.

$$FDII = \beta_0 + \beta_1 SEG + \beta_2 TROP + \beta_3 GR + \beta_4 EXC + \beta_5 INFR + \varepsilon \tag{2}$$

where $FDII$ = net "FDI inflow" which refer to the volume of inward capital inflows created by foreign investors in the host country. The economic meaning of FDI inflow is the FDI inflow as the percentage of GDP. Similarly, $SEG$ = sustainable economic growth, which refers to a pace of growth that can be sustained over the long term, especially for the next generations. The economic meaning of sustainable economic growth is GDP per capita, which measures USD using 2015 as a constant. The first regression coefficient "$\beta_1$" measures the rate of change in SEG. Similarly, the study also considered the control factors such as $TROP$ = Trade openness (%age of GDP), $GR$ = real growth rate (Annual %age), $EXC$ = exchange rate (official exchange rate), and $INFR$ = inflation rate (GDP deflator). The regression coefficient from $\beta_2 - \beta_5$ measures the rate of change in the above control variables. The estimation requires uniformity in terms of the unit of measurement for all the variables of the study to ensure linearity, e.g., the unit of measurement is in the percentage for all the variables except GDP per capital. For this purpose, a natural log for each variable is required that transforms the basic model into the following model.

$$LnFDII = \beta_0 + \beta_1 LnSEG + \beta_2 LnTROP + \beta_3 LnGR + \beta_4 LnEXC + \beta_5 LnINFR + \varepsilon \tag{3}$$

The basic panel data model for the study is as follows.

$$LnFDII_{it} = \beta_0 + \beta_1 LnSEG_{it} + \beta_2 LnTROP_{it} + \beta_3 LnGR_{it} + \beta_4 LnEXC_{it} + \beta_5 LnINFR_{it} + \mu_{it} \tag{4}$$

Additionally, the study also requires an estimation of the data using the fixed-effect model as follows:

$$LnFDII_{it} = \alpha_i + \beta_1 LnSEG_{it} + \beta_2 LnTROP_{it} + \beta_3 LnGR_{it} + \beta_4 LnEXC_{it} + \beta_5 LnINFR_{it} + \mu_{it} \tag{5}$$

Furthermore, the study also requires estimating the data using a random-effect model as follows:

$$LnFDII_{it} = \beta_0 + \beta_1 LnSEG_{it} + \beta_2 LnTROP_{it} + \beta_3 LnGR_{it} + \beta_4 LnEXC_{it} + \beta_5 LnINFR_{it} + (\mu_{it} + \alpha_i) \tag{6}$$

Finally, the study requires estimating the data using dynamic panel estimations methods; GMM uses the following equations.

## 4. Empirical Analysis

The research requires exploring the effect of sustainable economic growth on "FDI inflow" for the target population for the period of study along with trade openness, real growth, exchange rate, and inflation rate as the control variables. For this purpose, the study requires estimating two panels: "low-income" and "middle-income" economies separately from 1970 to 2021. The analysis of data requires some statistical and econometric procedures: comparative descriptive statistics, comparative stationarity testing, and panel regression estimations. The detail for each of these procedures is summarized in Tables 2–5.

**Table 2.** Comparative Summary Statistic.

|  | **Mean** | **STD** | **Min** | **Max** | **N** |
|---|---|---|---|---|---|
| | | **"Low-Income" Nations** | | | |
| *FDII* | 1.61 | 3.36 | −28.62 | 46.28 | 936.00 |
| *SEG* | 620.85 | 389.06 | 0.00 | 2133.21 | 936.00 |
| *TROP* | 47.54 | 23.74 | 0.00 | 140.86 | 936.00 |
| *RG* | 3.07 | 5.98 | −50.25 | 35.22 | 936.00 |
| *EXC* | 71.68 | 21.40 | 0.00 | 67.34 | 936.00 |
| *INFR* | 44.35 | 4.74 | −13.06 | 75.64 | 936.00 |
| | | **"Middle-Income" Nations** | | | |
| *FDII* | 1.96 | 3.26 | −55.23 | 39.25 | 2756.00 |
| *SEG* | 3306.44 | 2441.29 | 0.00 | 15,187.65 | 2756.00 |
| *TROP* | 63.44 | 39.67 | 0.00 | 274.97 | 2756.00 |
| *RG* | 3.88 | 5.54 | −64.05 | 57.82 | 2756.00 |
| *EXC* | 444.03 | 237.84 | 0.00 | 420.00 | 2756.00 |
| *INFR* | 32.75 | 22.21 | −16.12 | 49.38 | 2756.00 |

**Table 3.** Comparative Panel Unit Root Testing.

| | **Variables** | **IPS** | | **LLC** | |
|---|---|---|---|---|---|
| | | **Level** | **Δ** | **Level** | **Δ** |
| *"Low-Income" Economies* | *FDII* | −3.1190 *** | −5.2963 *** | −4.8759 *** | −8.6318 *** |
| | *SEG* | −2.5229 | −4.4678 *** | −2.7684 | −6.3767 *** |
| | *TROP* | −5.2570 *** | −6.2203 *** | −4.6024 *** | −9.8396 *** |
| | *RG* | −4.1539 *** | −7.8612 *** | −5.4138 *** | −6.6267 *** |
| | *EXC* | −2.028 | −3.016 *** | −2.2636 | −5.8564 *** |
| | *INFR* | −2.063 | −3.605 *** | −5.5126 *** | −8.1391 *** |
| *"Middle-Income" Economies* | *FDII* | −4.3345 *** | −10.3692 *** | −8.9578 *** | −33.8136 *** |
| | *SEG* | −1.7192 | −5.8764 *** | −1.4867 | −16.7673 *** |
| | *TROP* | −6.0752 *** | −7.0322 *** | −3.4260 *** | −22.6938 *** |
| | *RG* | −5.9351 *** | −11.2168 *** | −15.8314 *** | −36.7626 *** |
| | *EXC* | −2.280 | −4.610 *** | −1.6632 | −20.4658 *** |
| | *INFR* | −2.360 | −4.560 *** | −9.6215 *** | −32.1931 *** |

*** Significance at 1% level.

A comparative summary of the variables of the study for both panels, "low-income" and "middle-income" economies for the period of study, is reported in Table 2. The "FDI inflow" as a percentage of GDP indicates an average value of 1.61 for "low-income" economies and 1.96 for "middle-income" economies, respectively. Additionally, sustainable economic growth as GDP per capita (constant 2015) reports an average value of 620.85 for "low-income" economies and 3306.44 for "middle-income" economies, respectively. Similarly, trade openness as a percentage of GDP indicates an average value of 47.54 for "low-income" economies and 63.44 for "high-income" economies, respectively. Likewise, the real growth rate as GDP annual percentage indicates an average value of 3.07 for "low-income" economies and 3.88 for "middle-income" economies.

**Table 4.** Panel regression analysis for "low-income" economies.

| Variables | Static Panel (Robust Estimates) | | | | | Dynamic Panel (GMM) | |
|---|---|---|---|---|---|---|---|
| | FE | RE | OLS | DKSE | FGLS | Difference | System |
| L.FDII | - | - | - | - | - | 0.512 *** | 0.647 *** |
| | - | - | - | - | - | (0.0656) | (0.0369) |
| SEG | 0.347 *** | 0.296 *** | 0.0514 | 0.0514 | 0.0514 *** | 0.188 ** | 0.0449 ** |
| | (0.0987) | (0.0622) | (0.0393) | (0.0754) | (0.0154) | (0.0835) | (0.0190) |
| TROP | 0.118 *** | 0.092 *** | 0.138 *** | 0.138 *** | 0.138 ** | 0.0021 ** | 0.049 *** |
| | (0.017) | (0.012) | (0.0523) | (0.066) | (0.0622) | (0.0007) | (0.0065) |
| RG | 0.0512 | 0.0618 | 0.121 ** | 0.121 * | 0.121 ** | 0.0482 | 0.0681 * |
| | (0.0534) | (0.0541) | (0.0530) | (0.0656) | (0.0525) | (0.0369) | (0.0388) |
| EXC | 0.118 *** | 0.108 *** | 0.0325 *** | 0.0325 *** | 0.0325 *** | 0.0650 *** | 0.0142 |
| | (0.0220) | (0.0212) | (0.0113) | (0.0118) | (0.0117) | (0.0200) | (0.0118) |
| INFR | −0.0340 | −0.0341 | −0.0234 | −0.0234 | −0.0234 | −0.0169 | −0.00675 |
| | (0.0841) | (0.0798) | (0.0400) | (0.0490) | (0.0395) | (0.0418) | (0.0345) |
| Constant | −2.485 *** | −2.248 *** | −1.430 *** | −1.430 *** | −1.430 *** | - | −0.505 * |
| | (0.580) | (0.624) | (0.233) | (0.304) | (0.262) | - | (0.268) |
| Observations | 936 | 936 | 936 | 936 | 936 | 900 | 918 |
| Countries | 18 | 18 | 18 | 18 | 18 | 18 | 18 |
| $R^2$-Within | 0.488 | 0.487 | - | - | - | - | - |
| $R^2$-Between | 0.167 | 0.164 | 0.163 | 0.163 | - | - | - |
| $R^2$-Overall | 0.527 | 0.529 | - | - | - | - | - |
| Prob > F | 0.0000 | 0.0000 | 0.0000 | 0.0202 | 0.0000 | 0.000 | 0.000 |
| No of Instruments | - | - | - | - | - | 752 | 911 |
| **Diagnostic Tests** | | | | | | | |
| Hausman Test: Prob > chi2　0.0000 | | | | | | | |

| | | |
|---|---|---|
| ● Cross-Sectional Dependence (Pesaran's test) | 0.0000 | |
| ● Groupwise heteroskedasticity (Modified Wald test): Prob > chi2 | | 0.0000 |
| ● Autocorrelation in panel data (Wooldridge test): Prob > F | | 0.0166 |
| ● AR (I): Pr > z | 0.000 | 0.001 |
| ● AR (II): Pr > z | 0.086 | 0.089 |
| ● Sargan test: Prob > chi2 | 0.833 | 0.999 |
| ● Hansen test: Prob > chi2 | 1.000 | 1.000 |

Robust standard errors in parentheses; *** $p < 0.01$, ** $p < 0.05$, * $p < 0.1$.

The exchange rate, on the other hand, reports an average value of 71.68 for "low-income" economies and 444.03 for "middle-income" economies, respectively. Finally, the average value of inflation in the case of "low-income" economies is 44.35 and for "middle-income" economies is 32.75, respectively.

The summarized findings from descriptive statistics indicate that the average values of FDI, SEG, TROP, RG, and exchange rate are higher in the case of "middle-income" economies, while the inflation rate is higher in the case of "low-income" economies.

Table 3 reports the comparative panel stationarity testing using first-generation unit root testing procedures: Im–Pesaran–Shin (IPS) as introduced by [45] and Levin-Lin-Chu (LLC) as introduced by [46]. The test assumes a null hypothesis of unit root, the rejection of which confirms the stationarity of a particular variable. Table 3 indicates the stationarity testing using both methods at the level with the trend and at first difference (Δ) with the trend. The IPS and LLC methods indicate stationarity of FDII, TROP, and RG at the level for "low-income" economies as well as "middle-income" economies. Similarly, both methods indicate the stationarity of all the variables at first difference. The stationarity testing validates the estimations of panels using static panel techniques of fixed, random, and pooled OLS along with diagnostic tests such as [47] for choosing between fixed- or random-effect estimations, robust standard error procedures such as [48] in the presence of cross-sectional dependence, feasible generalized least squares by [49], difference and system GMM by [50], and, finally, panel-corrected standard error (PCSE) by [51].

**Table 5.** Panel regression analysis for "middle-income" economies.

| Variables | Static Panel (Robust Estimates) | | | | | Dynamic Panel (GMM) | |
|---|---|---|---|---|---|---|---|
| | FE | RE | OLS | DKSE | PCSE | Difference | System |
| L.FDII | - | - | - | - | - | 0.561 *** | 0.7274 |
| | - | - | - | - | - | (0.0301) | (0.0196) |
| SEG | 0.179 ** | 0.209 * | 0.387 *** | 0.387 *** | 0.387 *** | 0.00632 | 0.0986 *** |
| | (0.083) | (0.115) | (0.0468) | (0.0558) | (0.0406) | (0.0877) | (0.0367) |
| TROP | 0.0983 | 0.0866 | 0.0321 | 0.0321 | 0.0321 | 0.103 | 0.0122 |
| | (0.0752) | (0.0729) | (0.0323) | (0.0341) | (0.0295) | (0.0633) | (0.0261) |
| RG | 0.113 *** | 0.103 *** | 0.0132 | 0.0132 | 0.0132 | 0.0320 | 0.0037 |
| | (0.0333) | (0.0347) | (0.0384) | (0.0604) | (0.0401) | (0.0346) | (0.0296) |
| EXC | 0.150 *** | 0.136 *** | 0.0295 *** | 0.0295 *** | 0.0295 ** | 0.166 *** | 0.0113 ** |
| | (0.0337) | (0.0309) | (0.00748) | (0.00935) | (0.0121) | (0.0299) | (0.0041) |
| INFR | −0.105 ** | −0.100 ** | −0.102 *** | −0.102 *** | −0.102 *** | −0.0217 | −0.0142 *** |
| | (0.0471) | (0.0468) | (0.0234) | (0.0233) | (0.0266) | (0.0354) | (0.009) |
| Constant | −2.044 ** | −2.205 ** | −3.056 *** | −3.056 *** | −3.056 *** | - | −0.8050 *** |
| | (0.847) | (0.904) | (0.379) | (0.534) | (0.352) | - | (0.2843) |
| Observations | 2756 | 2756 | 2756 | 2756 | 2756 | 2650 | 2703 |
| Countries | 53 | 53 | 53 | 53 | 53 | 53 | 53 |
| $R^2$-Within | 0.520 | 0.501 | - | - | - | - | - |
| $R^2$-Between | 0.045 | 0.010 | 0.251 | 0.251 | 0.251 | - | - |
| $R^2$-Overall | 0.098 | 0.122 | - | - | - | - | - |
| Prob > F | 0.0000 | 0.0000 | 0.0000 | 0.0000 | 0.0000 | 0.000 | 0.000 |
| No of Instruments | - | - | - | - | - | 1280 | 2536 |
| **Diagnostic Tests** | | | | | | | |
| • Hausman Test: Prob > chi2 | 0.0000 | | | | | | |
| • Cross Sectional Dependence (Pesaran's test) | | | 0.0000 | | | | |
| • Groupwise heteroskedasticity (Modified Wald test): Prob > chi2 | | | | | 0.0000 | | |
| • Autocorrelation in panel data (Wooldridge test): Prob > F | | | | | 0.0058 | | |
| • AR (I): Pr > z | | | | | | 0.000 | 0.000 |
| • AR (II): Pr > z | | | | | | 0.057 | 0.064 |
| • Sargan test: Prob > chi2 | | | | | | 0.361 | 1.000 |
| • Hansen test: Prob > chi2 | | | | | | 1.000 | 1.000 |

Robust standard errors in parentheses; *** $p < 0.01$, ** $p < 0.05$, * $p < 0.1$.

Table 4 reports the statistic and dynamic panel estimations results for "low-income" economies. The table indicates that sustainable economic growth is statistically significant in increasing the "FDI inflow" in the target population during the period of study using fixed-effect, random-effect, FGLS, one-step difference GMM, and one-step system GMM techniques. The significance of the Hausman specification test confirms the validity of robust fixed-effect estimations using the static panel method. The coefficient value for sustainable economic growth indicates that "FDI inflow" for the target population is strongly enhanced by 0.347 by enhancing one unit in sustainable growth. The positive link between sustainable growth and "FDI inflow" in the "low-income" economies for the period of study accepts the first hypothesis. This finding is consistent with similar positive results from some studies [1,6,14,17,19,25–27,43]. One possible interpretation for this positive relationship is that long term and sustainable economic growth encourage foreign investors to enhance their confidence in the host country for making their investment inflow.

The table also reported that trade openness strongly increased the level of "FDI inflow" in "low-income" economies using statistic and dynamic panel regression models. The coefficient value of trade openness using the fixed-effect model indicates that a one percent increase in trade openness will strongly enhance the "FDI inflow" by 11.8% in the target population. The positive link between trade openness and "FDI inflow" accepts the second hypothesis. The positive relationship between the said variables is verified from the similar findings of some studies [12,33–36]. The possible interpretation for the positive relationship between trade openness and "FDI inflow" is that the trade openness between the two

different nations enhances the confidence of foreign investors to invest more in the host country and enhances the level of "FDI inflow" for the host nation. Table 4 reports the real growth being the insignificant influencer for "FDI inflow" for the target population during the period of study. The insignificant and positive relationship between economic growth and "FDI inflow" rejects the third hypothesis. The finding is verified by some important studies [4,12,39–41]. The possible interpretation for this insignificant and positive relationship between real growth and "FDI inflow" is that the real growth in "low-income" economies is always less than their projected target due to financing, technological, and other economic limitations due to which these economies were not able to attract much FDI based on their real growth rate.

Furthermore, the table reports that the exchange rate is positively linked with the "FDI inflow" using the majority of estimations methods using statistic panels and dynamic panels except for the one-step system GMM where it is not significant. The coefficient value using fixed-effect estimations between the exchange rate and "FDI inflow" is 0.118, which indicates that a one unit increase in the exchange rate will strongly enhance the FDI level by 0.118. The positive link between the said variables accepted the fourth hypothesis. The finding is also verified by some of the important literature evidence [8,12,39,42]. The possible interpretation for this positive relationship between the above-said variables is that the increase in exchange rate enhances the confidence of foreign investors for their investment to increase in value in terms of the host country's currency, therefore, the investors enhance their investment inflow in the host nation. Finally, the table reports an insignificant and negative impact of the inflation rate on "FDI inflow" for "low-income" economies for the period of study using all the estimations methods of static and dynamic panels. The insignificant and negative relationship between inflation rate and "FDI inflow" rejects the fifth hypothesis. The insignificant relationship of the above said variables are verified by similar results from some important studies [12,39,40]. The possible reason for this insignificant and negative impact is that the average inflation rate is higher in the case of "low-income" economies while the average "FDI inflow" is low as compared to "middle-income" economies. Secondly, inflation has no potential impact on "FDI inflow" in the case of "low-income" economies.

Table 5 reports the statistic and dynamic panel estimations results for "middle-income" economies. The table indicates that sustainable economic growth is statistically significant in increasing the "FDI inflow" in the target population during the period of study using fixed-effect, random-effect, PCSE, and one-step system GMM techniques. The significance of the Hausman specification test confirms the validity of robust fixed-effect estimations using the static panel method. The coefficient value for sustainable economic growth indicates that "FDI inflow" for the target population is strongly enhanced by 0.179 by enhancing one unit in sustainable growth. The positive link between sustainable growth and "FDI inflow" in the "middle-income" economies for the period of study accepts the first hypothesis. This finding is consistent with similar positive results from some important studies [3,6,17,19,27,43]. One possible interpretation for this positive relationship is that long term and sustainable economic growth encourage foreign investors to enhance their confidence in host countries in "middle-income" economies for making their investment inflow.

However, the table reported trade openness has an insignificant and positive impact on "FDI inflow" in "middle-income" economies using statistic and dynamic panel regression models. This finding rejects the second hypothesis. The insignificant but positive relationship between the said variables is verified from similar findings of some important studies [7,10,11,18,25,29,30,32]. The possible interpretation for this insignificant relationship between trade openness and "FDI inflow" in "middle-income" economies is that the trade openness does not potentially contribute to enhancing the foreign investor's confidence in making their investment in the host country in "middle-income" economies. Table 4 reports that real growth strongly enhances the "FDI inflow" for the target population during the period of study for "middle-income" economies using fixed-effect and random-effect esti-

mations only. The coefficient value using the fixed-effect method is 0.113, which indicates that a one unit increase in real growth rate can strongly enhance the "FDI inflow" by 0.113 in "middle-income" economies. The finding is verified by some studies [9–11,29,37,38]. The possible interpretation of a strong and positive relationship between real growth and "FDI inflow" is that the real growth in "middle-income" economies is always greater than their projected target due to financing, technological, and other economic advantages as compared to "low-income" economies that can attract much FDI based on their real growth rate.

Furthermore, the table reports that the exchange rate is positively and significantly linked with "FDI inflow" in "middle-income" economies using all the estimations methods using statistic panels and dynamic panels. The coefficient value using fixed-effect estimations between exchange rate and "FDI inflow" is 0.150, which indicates that a one unit increase in the exchange rate will strongly enhance the FDI level by 0.150. The positive link between the said variables accepted the fourth hypothesis. The finding is also verified by some of the literature evidence [10,11,29,42]. The possible interpretation for this positive relationship between the above-said variables is that the increase in exchange rate enhances the confidence of foreign investors for their investment to increase in value in terms of the host country's currency, therefore, the investors enhance their investment inflow in the host nation for "middle-income" economies. Finally, the table reports and highly significant and negative impacts of the inflation rate on "FDI inflow" for "middle-income" economies for the period of study using all the estimations methods of static and dynamic panels. The highly significant and negative relationship between inflation rate and "FDI inflow" accepts the fifth hypothesis. The significant relationship between above said variables are verified in some studies [4,7,11,29,42]. The possible reason for this highly significant and negative impact is that the average inflation rate is lower in the case of "middle-income" economies while the average "FDI inflow" is higher as compared to "low-income" economies. Secondly, inflation has a potential impact on "FDI inflow" in the case of "middle-income" economies.

## 5. Conclusions

The present research aims to explore the impact of sustainable economic growth on "FDI inflow" using comparative panel econometrics on two panels: "low-income" and "middle-income" economies from 1970 to 2021. The study considered "middle-income" and "low-income" economies because these countries need to enhance their level of FDI inflow to strengthen their income and increase their gross national income (GNI). As "high-income" groups are already earning the highest level of gross national income and do not need to bother about their level of "FDI inflow", the study considered only "middle-income" and "low-income" economies.

The findings revealed that sustainable growth enhances "FDI inflow" in "low-income" economies for the period of study. Additionally, trade openness, and exchange rate also potentially enhance the "FDI inflow" for the target population. However, inflation having a negative impact and the real growth rate having a positive impact on "FDI inflow" was not significant for "low-income" economies during the period of study. Similarly, sustainable growth also strongly enhances the "FDI inflow" in "middle-income" economies for the period of study. Likewise, the real growth rate and exchange rate are strongly enhanced while inflation strongly decreases the "FDI inflow" in the target population for the period of study. The trade openness, however, could not provide sufficient evidence to strongly enhance the "FDI inflow" in "middle-income" economies. The study confirms the application of the growth theory of FDI as developed by [13] and extended by [14], which provides a positive relationship between sustainable economic growth and "FDI inflow" for "middle-income" and "low-income" economies.

The analysis implies that the policymakers in "low-income", as well as "middle-income" economies, should maintain a long-run sustainable economic growth to attract more "FDI inflow" in their economies. The policymakers in "low-income" economies

should also facilitate foreign investors regarding the trade openness and exchange rate to ensure a positive inflow of FDI in their economies. Similarly, the policymakers in "middle-income" economies should consider their policies regarding the real growth rate and exchange rate as the contributory factors to enhancing the "FDI inflow" in their nations and inflation as the contributory factor to decreasing the "FDI inflow" for "middle-income" economies. The policymakers should change their FDI inflow attraction policy as per the implications of the study for target populations. The findings add the existing literature for the relationship between sustainable economic growth and FDI inflow as a comparison between "low-income" and "middle-income" economies. The role of sustainable economic growth was mainly ignored for attracting the FDI inflow in the target population as per the existing literature. The findings of the study are applicable and generalizable in the case of "middle-income" and "low-income" economies only.

Future researchers can potentially include some more control variables and consider the study for "high-income" groups.

**Author Contributions:** Conceptualization, M.A.H. and M.U.A.; Formal analysis, M.A.H. and M.U.A.; Investigation, S.M.R.S.; Methodology, M.A.H.; Software, S.M.R.S. and M.U.A.; Supervision, S.M.R.S.; Validation, S.M.R.S.; Visualization, S.M.R.S.; Writing—original draft, M.A.H.; Writing—review & editing, M.U.A. All authors have read and agreed to the published version of the manuscript.

**Funding:** This research received no external funding.

**Institutional Review Board Statement:** Not applicable.

**Informed Consent Statement:** Not applicable.

**Data Availability Statement:** The data for this study was obtained from the World Bank's data sources; world development indicators (WDI) using the annual frequency of 1970–2021.

**Conflicts of Interest:** The authors declare no conflict of interest.

## Appendix A

### List of countries included in the analysis

| Middle-Income Countries | | Low-Income Countries |
|---|---|---|
| 1. Algeria | 28. Kenya | |
| 2. Bangladesh | 29. Malaysia | |
| 3. Bolivia | 30. Mauritania | |
| 4. Botswana | 31. Mauritius | |
| 5. Brazil | 32. Mexico | 1. Benin |
| 6. Cameroon | 33. Morocco | 2. Burkina Faso |
| 7. Colombia | 34. Myanmar | 3. Burundi |
| 8. Congo, Rep. | 35. Nicaragua | 4. Central African Republic |
| 9. Costa Rica | 36. Nigeria | 5. Chad |
| 10. Côte d'Ivoire | 37. Pakistan | 6. Congo, Dem, Rep |
| 11. Dominican Republic | 38. Panama | 7. Gambia, |
| 12. Ecuador | 39. Papua New Guinea | 8. Haiti |
| 13. Egypt, Arab Rep. | 40. Paraguay | 9. Madagascar |
| 14. El Salvador | 41. Peru | 10. Malawi |
| 15. Eswatini | 42. Philippines | 11. Mali |
| 16. Fiji | 43. Senegal | 12. Niger |
| 17. Gabon | 44. Solomon Islands | 13. Rwanda |
| 18. Ghana | 45. South Africa | 14. Sierra Leone |
| 19. Guatemala | 46. Sri Lanka | 15. Somalia |
| 20. Guyana | 47. Sudan | 16. Togo |
| 21. Honduras | 48. Suriname | 17. Uganda |
| 22. India | 49. Syrian Arab Republic | 18. Zimbabwe |
| 23. Indonesia | 50. Thailand | |
| 24. Iran, Islamic Rep. | 51. Tunisia | |
| 25. Iraq | 52. Turkey | |
| 26. Jamaica | 53. Zambia | |
| 27. Jordan | | |

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
