# Peer review of "Sustainable Economic Growth and FDI Inflow: A Comparative Panel Econometric Analysis of Low-Income and Middle-Income Nations"

_sustainability, doi:10.3390/su142114321_

Round 1
Reviewer 1 Report
In this paper, the authors use econometric analysis method to investigate the relationship between economic growth and FDI inflow. There are some problems in this version:
1. Title: lower and middle income nations are different with low income and middle income nations.
2. Figure 1, the units should be added.
3. Page 3, the research objects and research questions are too similar, and they should not be numbered consecutively.
4. The serial number of the sections should be checked.
5. In one paper, we say sections, not chapters.
6. For the hypothesis development, you should use some theory and/or evidence to support your judgment. Only list some literature is not enough to support your judgment. There are literatures supporting positive or negative relationship, why you only support positive? We cannot judgment dependent on the quantity of related researches, but should check their theory, explanations, and evidence.
7. Only list large amounts of papers is not a proper way of citation. You should add some necessary comments.
8. There are five hypothesis. It seems like that they are rather parallel, why you say you only focus on the first one. In one paper, you should distinguish the primary and secondary.
9. What is the difference between sustainable economic growth and economic growth? Does per capita GDP is a good measure for sustainable?
10. Is real growth is a good name for growth rate?
11. CPI and percentage of GDP deflator are both commonly used measures for inflation, but they are not the same. In your research, GDP deflator maybe better than CPI.
12. Beside per capita GDP, other variables are the percentages, and usually would vary largely. Do you need to use the logarithm of these variables? The explanations of the regression coefficients are not so direct if you use logarithm of these percentages.
13. What are the units of per capita GDP and exchange rate?
14. For inflation rate, the maximum is 23773.13 which clearly seems abnormal. You should consider the influence of outliers. Besides, low income nations and middle income nations have the same largest inflation rate, is it true?
15. Table 3, what does ∆ mean?
16. The explanation of the regression coefficients should be consider the definition of the variables, the economic meaning and the effect of natural log.
17. The first paragraph of conclusion only reported how you do the research, not very informative for the readers. Just simply say what you have done, and focus on your findings and implications.
18. Why you only studied low income and middle income nations? Why high income groups were not included?
Author Response
Reviewer 1:
Comment 1: Title: lower and middle-income nations are different from low-income and middle-income nations.
Answer to Comment 1: Incorporated on Page 1, Line 3 in the main file.
Comment 2: In figure 1, the units should be added.
Answer to Comment 2: Incorporated on Page 1, line 38, and Page 2, line 44 in the main file
Comment 3: On page 3, the research objects and research questions are too similar, and they should not be numbered consecutively.
Answer to Comment 3: Incorporated on Page 3, lines 61-69. The research questions were removed due to their similarity with the research objectives.
Comment 4: The serial number of the sections should be checked.
Answer to Comment 4: Changes incorporated on Page 3 (line 81), Page 5 (line 193), Page 7 (line 272), and Page 13 (line 447)
Comment 5: In one paper, we say sections, not chapters.
Answer to Comment 5: Changes incorporated on page 3 (lines 77-79).
Comment 6: For the hypothesis development, you should use some theory and/or evidence to support your judgment. Only listing some literature is not enough to support your judgment. There are literature supporting positive or negative relationship, why do you only support positive ones? We cannot judgments dependent on the number of related research but should check their theory, explanations, and evidence.
Answer to Comment 6: Changes Incorporated on page 3 (lines 95-100).
Comment 7: Only listing large amounts of papers is not a proper way of citation. You should add some necessary comments.
Answer to Comment 7: Changes incorporated in literature review, page 3-4, lines 111-123, , Lines 131-136, 130-135, 137-143, 148-151, 154-160, 166-168, 170-176, and 181-189
Comment 8: There are five hypotheses. It seems like they are rather parallel, why do you say you only focus on the first one? In one paper, you should distinguish the primary and secondary.
Answer to Comment 8: Changes incorporated on page 2 (line 61 & 62), page 4 (line 125, 144-145, 162, 177-178, 191)
Comment 9: What is the difference between sustainable economic growth and economic growth? Does per capita GDP a good measure of sustainability?
Answer to Comment 9: Changes incorporated on page 5, lines 209-217.
Comment 10: Is real growth a good name for growth rate?
Answer to Comment 10: Changes incorporated on page 5, lines 219-221
Comment 11: CPI and percentage of GDP deflator are both commonly used measures for inflation, but they are not the same. In your research, the GDP deflator may be better than CPI.
Answer to Comment 11: Changes incorporated on Page 6, Table 1, and lines 226-227
Comment 12: Besides per capita GDP, other variables are the percentages, and usually would vary largely. Do you need to use the logarithm of these variables? The explanations of the regression coefficients are not so directly if you use the logarithm of these percentages.
Answer to Comment 12: Changes incorporated on Page 7, lines 264-267
Comment 13: What are the units of per capita GDP and exchange rate?
Answer to Comment 13: Changes incorporated on Page 6, Table 1, and lines 226-227
Comment 14: For the inflation rate, the maximum is 23773.13 which seems abnormal. You should consider the influence of outliers. Besides, low-income nations and middle-income nations have the same largest inflation rate, is it true?
Answer to Comment 14: Changes incorporated on page 8, table 2. The new values are incorporated after the adjustment of outliers. The outlier was removed and corrected values are indicated now.
Comment 15: Table 3, what does ∆ mean?
Answer to Comment 15: Changes incorporated on Page 8, line 309
Comment 16: The explanation of the regression coefficients should consider the definition of the variables, the economic meaning, and the effect of the natural log.
Answer to Comment 16: Changes incorporated page 7, lines 257-267
Comment 17: The first paragraph of the conclusion only reported how you do the research, not very informative for the readers. Just simply say what you have done, and focus on your findings and implications.
Answer to Comment 17: Changes incorporated on page 13. The unnecessary part from the first paragraph is removed. The conclusion now focuses on findings and implications.
Comment 18: Why you only studied low-income and middle-income nations? Why are high-income groups not included?
Answer to Comment 18: Changes incorporate on page 13, the first paragraph of conclusion. Line 453-458 added and updated.
Reviewer 2 Report
The article is of interest from the perspective of the long period of analysis and the selected groups of countries, considering the different behavior of FDI in the destination country, especially in the light of current challenges.
The paper must be improved from the perspective of personal contributions, considering the fact that the findings presented are already well known an proved by the specialists since decades ago, through numerous other studies (i.e. Zaman 2012, DOI 10.1016/S2212-5671(12) 00113-X, Vasile V 2020 DOI: 10.17605/OSF.IO/2ZU9J or many others) The authors must show us what the current research brings new, compared to the state of knowledge.
It is natural that research objectives should be transposed into research questions, and when there is a linearity between objective and question, then it is indicated to specify only one of them, as a way of empirical analysis of the general objective, already stated. I recommend giving up one of these, because, in fact, 1=3 and 2=4 (page 3)
A substantial revision of the writing in English is necessary, some expressions being inappropriate - for example "FDI inflow" is the financing made by foreign investors in any nation.
The concept of "nation" is inadequately used, not being synonymous with country!
Attention to the use of bibliographic sources! you must mention the initial source for the definition of sustainable development as long as it is not modified in essence, namely 1987, the United Nations Brundtland Report and not recent takeovers such as -Sial et al., 2022.
Attention to transient statements, which do not correspond to reality! What does "a limited number of studies still provided the insignificant yet positive impact of trade openness on "FDI inflow" for individual countries as well as on some panel studies" mean, as long as, a simple scientometric query on keywords only in the database WoS indicates an important number of studies, namely over 4400 works.
There are no complete specifications regarding the database used! The countries included in the two analysis groups are not indicated! Why 18 and 53? What were the sampling criteria, is representativeness ensured, what are the selection characteristics, etc. All mentions regarding the database and methodological details must be presented for the first time in the "methodology and database" section and not in the conclusions!
At the conclusions, it is necessary to indicate the added value of the obtained results and the elements of originality, the limits of the research. Appraisals of usefulness are too general. What are the recommendations for changing the FDI attraction policy resulting from the analysis?
Author Response
Reviewer 2:
Comment 1: The paper must be improved from the perspective of personal contributions, considering the fact that the findings presented are already well known an proved by the specialists since decades ago, through numerous other studies (i.e. Zaman 2012, DOI 10.1016/S2212-5671(12) 00113-X, Vasile V 2020 DOI: 10.17605/OSF.IO/2ZU9J or many others) The authors must show us what the current research brings new, compared to the state of knowledge.
Answer to Comment 1: Changes incorporated on Page 1, Line 21-23
Comment 2: Naturally, research objectives should be transposed into research questions, and when there is a linearity between objective and question, then it is indicated to specify only one of them, as a way of empirical analysis of the general objective, already stated. I recommend giving up one of these, because, in fact, 1=3 and 2=4 (page 3).
Answer to Comment 2: Changes incorporated on page 2-3. Research question were removed. Only research objectives were retained (line 66-69).
Comment 3: A substantial revision of the writing in English is necessary, some expressions being inappropriate - for example "FDI inflow" is the financing made by foreign investors in any nation.
Answer to Comment 3: Changes incorporated in the full article. Proofreading and professional English editing is done to make the article error-free in terms of grammar, sentence structure, and English language.
Comment 4: The concept of "nation" is inadequately used, not being synonymous with country!
Answer to Comment 4: Changes incorporated on Page 3, line 70-73
Comment 5: Attention to the use of bibliographic sources! you must mention the initial source for the definition of sustainable development as long as it is not modified in essence, namely 1987, the United Nations Brundtland Report and not recent takeovers such as -Sial et al., 2022.
Answer to Comment 5: Changes Incorporated on page 3 line 82-83
Comment 6: Attention to transient statements, which do not correspond to reality! What does "a limited number of studies still provided the insignificant yet positive impact of trade openness on "FDI inflow" for individual countries as well as on some panel studies" mean, as long as, a simple scientometric query on keywords only in the database WoS indicates an important number of studies, namely over 4400 works.
Answer to Comment 6: Changes incorporated on Page 9 line 353-354, 374-376, Page 11, line 403-405
Comment 7: There are no complete specifications regarding the database used! The countries included in the two analysis groups are not indicated! Why 18 and 53? What were the sampling criteria, is representativeness ensured, what are the selection characteristics, etc. All mentions regarding the database and methodological details must be presented for the first time in the "methodology and database" section and not in the conclusions!
Answer to Comment 7: Changes incorporated on page 5 under data and methodlogy section on line 199-207. Additionally, further changes incorporated by removing the unnessary part of methodology on page 13 in conclusion section. Additionally, a table included for sample description as appendix-1.
Comment 8: At the conclusions, it is necessary to indicate the added value of the obtained results and the elements of originality, the limits of the research. Appraisals of usefulness are too general. What are the recommendations for changing the FDI attraction policy resulting from the analysis?
Answer to Comment 8: Changes Incorporated for value addition, originality, and limits of research on page 13, conclusion section, lines 481-485. Additionally, the changes incorporated for the recommendation of changing the FDI attraction policy on page 13 and lines 485-486
Round 2
Reviewer 1 Report
It is much improved now.
Reviewer 2 Report
No additional comments